# Exploring with Sticky Mittens: Reinforcement Learning with Expert Interventions via Option Templates

**Souradeep Dutta[1], Kaustubh Sridhar[1], Osbert Bastani[1], Edgar Dobriban[1], James Weimer[2],**

**Insup Lee[1], Julia Parish-Morris[3]**

[1]University of Pennsylvania, [2]Vanderbilt University, [3]Children's Hospital of Philadelphia

**Abstract:** Long horizon robot learning tasks with sparse rewards pose a significant challenge for current reinforcement learning algorithms. A key feature enabling humans to learn challenging control tasks is that they often receive expert intervention that enables them to understand the high-level structure of the task before mastering low-level control actions. We propose a framework for leveraging expert intervention to solve long-horizon reinforcement learning tasks. We consider *option templates*, which are specifications encoding a potential option that can be trained using reinforcement learning. We formulate expert intervention as allowing the agent to execute option templates before learning an implementation. This enables them to use an option, before committing costly resources to learning it. We evaluate our approach on three challenging reinforcement learning problems, showing that it outperforms state-of-the-art approaches by two orders of magnitude. Videos of trained agents and our code can be found at: https://sites.google.com/view/stickymittens

**Keywords:** Sample-Efficient Reinforcement Learning, Expert Intervention, Options, Planning with Primitives

## 1 Introduction

Reinforcement learning is an effective tool to solve difficult tasks such as robot planning and loco-motion [1] but exploration is still a challenge. *Options* are an RL tool to circumvent this problem [2]. Designed to achieve intermediate subgoals. For instance, in robot grasping tasks, an option might enable the robot to grasp a block, which is a subgoal needed to build a tower out of blocks. The goal is to learn a policy mapping each state to an option, instead of a concrete action to take.

When learning to perform complex visual-motor skills, humans often rely on expert interventions to help them escape these challenging reward plateaus. For instance, the *sticky-mittens* experiment [3] considers infants who have not yet learned to grasp objects. They give a subset of these infants mittens covered with Velcro hooks and allow them to play with toys fitted with Velcro loops, making it significantly easier for them to grasp these toys. Even if the Velcro is taken away, these babies learn how to grasp objects significantly faster than infants not exposed to this experience. In other words, enabling infants to explore unreachable parts of the state space helps guide them towards skills that are worth learning. This is a well known phenomenon in developmental psychology, which extends beyond fine motor skills.

In this paper, we design an RL algorithm based on the idea from the sticky-mittens experiment. The agent has access to an alternative Markov Decision Process (MDP) where the agent can leap multiple states without first learning a policy to do so in the original MDP. We term such a jumping mechanism an *option template*. Option templates are described using a set of initial states and a set of final states. The idea of providing external help in the learning phase is referred to as *primitives* or *skills* in literature. It offers a practical way to speed-up learning in a realistic setting. For instance,

using parameterized action spaces for RoboCup in [4], stitching independent behaviors in [5, 6] and providing action primitives in [7, 8, 9] to mention a few.

In the more typical RL framework the agent first uses reinforcement learning to train an option to implement the specification of each option template. The issue with this strategy is that, in RL environments with large state spaces the options learnt need to be of a fairly generic nature, in order to be useful. Which is hard without a knowledge of the state distribution, where the options will be invoked. Using option templates we decouple the implementation of an option from its utility.

In more detail, our algorithm performs an alternating search over policies: at each iteration, it first optimizes the high-level policy over the current option templates, and then learns options to implement the option templates. In our experiments, we demonstrate that by leveraging option templates, our algorithm can achieve orders of magnitude reduction in sample complexity compared to the typical strategy of learning the options upfront and then planning with these learned options.

**Contributions:** Our contributions are: (1) an RL algorithm that leverages option-templates, and (2) an empirical comparison with state-of-the-art RL techniques on three challenging environments, demonstrating order-of-magnitudes reduction in sample complexity.

## 2 Background

We consider the RL setting where an agent interacts with an environment modeled as an MDP [10].

**Definition 2.1 (MDP).** A *Markov Decision Process (MDP)* is a tuple $\mathcal{E} = (\mathcal{S}, \mathcal{A}, \mathcal{P}, \mathcal{R}, \gamma, \mathcal{I})$, where $\mathcal{S} \subseteq \mathbb{R}^n$ is the set of states, $\mathcal{A} \subseteq \mathbb{R}^m$ is the set of actions, $\mathcal{P}(s'|s, a)$ is the probability of transitioning from state $s$ to $s'$ upon taking action $a$, $\mathcal{R}(s, a)$ is the reward accrued in state $s$ upon taking action $a$, $\gamma \in [0, 1)$ is the discount factor, and $\mathcal{I}$ is the initial state distribution.

The value of a state for policy $\pi$ is the expected return $\mathbf{R}_\pi(s_0)$ starting from state $s_0$, while executing the policy $\pi$ at every step, $a_t = \pi(s_t)$. The optimal policy $\pi^*$ maximizes the reward starting from the initial state distribution–i.e., $\pi^* = \arg \max V^{\mathcal{I}}(\pi)$, where $V^{\mathcal{I}}(\pi) = \mathbb{E}_{s_0 \in \mathcal{I}}[\mathbf{R}_\pi(s_0)]$. Semi-MDPs (SMDPs) extend this framework by allowing temporally extended actions—i.e., policies executed over multiple steps—called *options* [2].

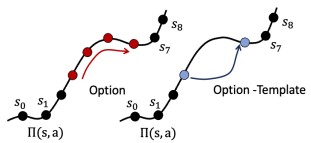

Figure 1: Option vs Option Template.

**Definition 2.2 (Option).** An *option* $o$ is a tuple $o := (I, T, \pi)$, where $I \subseteq S$ are initial states, $T : \mathcal{S} \rightarrow [0, 1]$ the termination condition, and $\pi : \mathcal{S} \rightarrow \mathcal{A}$ is a policy.

For now, assume that we have a set $\mathcal{O}$ of options available for an MDP. Intuitively, options constrain the search space over policies, thereby guiding exploration performed by the agent towards more promising action sequences. We now consider policies $\Pi : \mathcal{S} \rightarrow \mathcal{O}$ that map a state $s \in I$ (i.e., $s$ is a valid initial state for $o$) to an option $o = \Pi(s)$. The agent uses option policy $\pi$ until the termination condition holds. More precisely, after taking action $a = \pi(s)$ and transitioning to state $s' \sim \mathcal{P}(\cdot|s, a)$, it stops using $\pi$ with probability $T(s_{t+1})$ and chooses another option $o' = \Pi(s')$; otherwise, it continues using $\pi$.

Suppose that an option $o$ is invoked at time $t$ in state $s_t$, and the system follows a $k$ step trajectory $\mathsf{Tr}^o(s_t, k) = (s_t, a_t, r_t, s_{t+1}, \dots, s_{t+k})$. We consider policies such that for $t \leq i < t + k$, the action $a_i$ depends on the entire history $\mathsf{Tr}^o(s_t, i)$ from $t$ until $t + i$. This renders the behavior non-Markovian. Denoting the set of all such state-action trajectories as $H$, we can allow option policies $\pi : H \rightarrow \mathcal{A}$, and termination conditions $T : H \rightarrow [0, 1]$ to be semi-Markovian. Now, we recall the definition of Semi-MDPs as an extension of MDPs:

**Definition 2.3 (Semi-MDP,** Sutton et al. [2]**).** A semi-MDP is a tuple $(\mathcal{S}, \mathcal{O}, \mathcal{P}, \mathcal{R}, \gamma, \mathcal{I})$, where $\mathcal{S}$ is a set of states, $\mathcal{O}$ is a set of semi-Markovian options, $\mathcal{P}$ is the transition probability between states, $\mathcal{R}$ is the reward function, $\gamma \in [0, 1)$ is the discount factor, and $\mathcal{I}$ is the initial state distribution.

The state-prediction part of the model for taking an option $o$, in state $s \in I$ and transitioning to $s'$, is given by Sutton et al. [2]: $P(s'|s, o) = \sum_{k=1}^{\infty} \gamma^k p(s', s, k)$, where $p(s', s, k)$ is the probability that the option terminates in $s'$ after $k$ steps from the time the option is invoked. The option-value form of standard value iteration is, for $s \in \mathcal{S}$ and $o \in \mathcal{O}$, [2, eq. 12], [11] $Q^*_{\mathcal{O}}(s, o) = R(s, o) + \sum_{s' \in S} P(s'|s, o) \max_{o' \in \mathcal{O}} Q^*_{\mathcal{O}}(s', o')$, where $s'$ is the state the system reaches after executing option $o$ starting from state $s$. SMDP value learning updates the $Q$-values at the end of each

option termination as [2, p. 195]: $Q^{i+1}(s, o) \leftarrow Q^i(s, o) + \alpha[r_c + \gamma^k \max_{o' \in \mathcal{O}} Q^i(s', o') - Q^i(s, o)]$, for similar definitions of $k$, $s$, $s'$, with $r_c$ being the cumulative discounted reward over the time horizon, and $\alpha \in (0, 1)$ the learning rate. Choosing a function approximator parameterized by $\theta$ to represent $Q(s, o; \theta)$, with a loss given by : $L(\theta_i) = [r + \gamma^k \max_{o' \in O} Q^i(s', o'; \theta_i) - Q^i(s, o; \theta_i)]^2$, allows one to use standard Q-learning [12] methods to train with options.

# 3 Learning with Option Templates

## 3.1 Option Templates

Motivated by the sticky-mittens experiment, we consider an environment where in certain states, the agent can call a "help switch", called an *option template*, that immediately transitions it from a state $s_t$ to a different state $s_{t+k}$; this transition captures the desired result of executing an (unimplemented) option. We denote a trajectory of this MDP starting from state $s_t$ under policy $\pi$ by $\mathsf{Tr}^\pi(s_t)$.

**Definition 3.1 (Option template).** Option template $a_o$ is a tuple $(I, T, P)$, where $I \subseteq \mathcal{S}$ is the initial states, $T : \mathcal{S} \to [0, 1]$ is the termination condition, and $P$ is a distribution over states $s$ such that $T(s) = 1$.

An option template similar to an option, shifts control to the expert until the termination condition is reached; this termination condition $T$ is intended to capture the satisfaction of some sub-goal along with a timeout mechanism. Assume the termination condition at time $t + k$ of an option template taken in state $s_t$ is of the form $T(s_t, k) = \mathbf{F}_o(s_{t+k}) \vee (k > k^*)$, where the *function* $\mathbf{F}_o : \mathcal{S} \to \{0, 1\}$ captures visitation of some key states, and $k^*$ is an upper limit on the number of steps.

We denote the set (or subset) of option templates by $\mathcal{A}_\mathcal{O}$. By the end of training, the agent must learn a policy to implement each option template that it uses (i.e., the final policy cannot depend on option templates). To ensure feasiblity, we assume there is an a priori unknown policy $\pi_o$ such that $\mathsf{Tr}^{\pi_o}(s_t)$ satisfies $T$ (denoted as $\mathsf{Tr}^{\pi_o}(s_t) \models T$) with probability at least $1 - \delta$, for some hyperparameter $\delta \in [0, 1)$. Formally, the probability of success of a given policy $\pi_o$ can be expressed as $\mathbb{P}((\pi_o, I, \mathcal{E}) \models T) \geq (1 - \delta)$—i.e., the trajectory starting from a state in $I$ under the control of $\pi_o$ in environment $\mathcal{E}$ satisfies $T$ with probability at least $1 - \delta$. This success probability can be estimated by empirical rollouts of $\pi_o$.

We organize option-templates in a hierarchical fashion depending on degrees of help. Option templates that traverse longer sequences of transitions allow faster learning due to fewer decisions. Thus, we consider a sequence of environments $\mathcal{E}^0, \mathcal{E}^1, \ldots, \mathcal{E}^{n-1} = \mathcal{E}$, equipped with option templates of varying capabilities organized in a hierarchical fashion.

**Definition 3.2 (Environment Level $\mathcal{E}^l$).** An environment $\mathcal{E}^l$ at learning level $l$ is the original environment $\mathcal{E}$ with the primitive actions $\mathcal{A}$ replaced by option templates $\mathcal{A}_\mathcal{O}^l$.

The goal of the levels is to gradually increase difficulty of the learning task while guiding the agent at each level in a way that is similar to curriculum learning [13]. In particular, learning longer timescale option templates typically precedes learning shorter ones.

**Assumption 3.3 (Realizability).** For any option template $a_o \in \mathcal{A}_\mathcal{O}^l$ at level $l$, given by $(I, T, P)$, there is a policy $\pi^{a_o} : \mathcal{S} \to \mathcal{A}_\mathcal{O}^{l+1}$ at the following level $l + 1$ such that for any $s_0 \in I$, $\mathsf{Tr}^o(s_0) \models T$ with probability at least $1 - \delta$.

That is, any option template in the current level is realizable by a policy in the following level.

## 3.2 Learning with Option Templates

Our algorithm for RL with option templates is presented in Algorithm 1.

**Algorithm 1** Learning with Option Templates

**Input:** Environments $[\mathcal{E}^0, \mathcal{E}^1, \ldots, \mathcal{E}^{n-1}]$
**Output:** A set of options $\mathcal{D} = \{o^q \mid q \in Q\}$ that provides an implementation of each option template $a_o^q$, where $q \in Q$ indexes the option templates.

1: Initialize $\mathcal{D} = \{\}$
2: **for** level $l \in \{1, \ldots, n\}$ **do**
3:     Let $\mathcal{A}_o^{l-1}$ be the set of option templates on level $l - 1$
4:     Build reward functions $\{\mathfrak{R}_q \mid a_o^q \in \mathcal{A}_o^{l-1}\}$
5:     **for** $a_o^q \in \mathcal{A}_o^{l-1}$ **do**
6:         $\pi_o^q =$ LearnOptionPolicy$(\mathcal{E}^l, \mathfrak{R}_q, a_o^q, \mathcal{D})$
7:         $o^q := (a_o^q, \pi_o^q)$
8:         $\mathcal{D} = \mathcal{D} \cup \{o^q\}$
9:     **end for**
10: **end for**
**output** $\mathcal{D}$

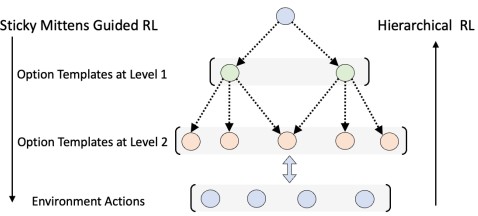

Figure 2: Visualization of learning with option templates.

**Algorithm 2** Learn Option Policy

**Input:** Environment $\mathcal{E}$, Reward Function $\mathfrak{R}$, Option template $a_o = (I, T, P)$, Set $\mathcal{D}$ of options
**Parameter:** Threshold $\delta$, Max Episodes $L$
**Output:** Policy $\pi_o$ as an implementation of $a_o$

1: Initialize policy $\pi_o$
2: **for** episode $= 1, \ldots, L$ **do**
3:     *ExecutionStack* $= []$
4:     $o \leftarrow$ Top-level option from $\mathcal{D}$ (or $a_o$ if $\mathcal{D} = \varnothing$)
5:     Push $o$ onto *ExecutionStack*
6:     Sample initial state $s \sim I$
7:     **while** not done **do**
8:         $o \leftarrow$ Pull option from *ExecutionStack*
9:         $o' \leftarrow \pi_o(s)$
10:         **if** IsOption$(o', a_o)$ **then**
11:             $(\pi_o, s) \leftarrow$ StepAndTrain$(\mathcal{E}, \mathfrak{R}, \pi_o, s)$
12:         **else if** $\neg$HasImplementation$(o', \mathcal{D})$ **then**
13:             $s \leftarrow$ Teleport$(o', s)$
14:         **else**
15:             Push $o'$ to *ExecutionStack*
16:         **end if**
17:     **end while**
18:     Exit loop if AvgReward $\geq 1 - \delta$
19: **end for**
**output** $\pi_o$

Intuitively, at each iteration, it "flattens" the option templates used on the previous level $l - 1$ based on the ones available at the current level $l$. At each level $l$, we first design reward functions $\mathfrak{R}_q$ for each option template $a_o^q \in \mathcal{A}_o^{l-1}$ (Line 4); this reward function is later used to learn a policy $\pi_o^q$ implementing $a_o^q$. For the base case $l = 1$, we take the reward function $\mathfrak{R}_q$ to be the reward function for the original environment $\mathcal{E}$; thus, learning the policy for the option-template at this level amounts to accomplishing the goals of $\mathcal{E}$ using the (unique) option template $a_o^q \in \mathcal{A}_o^0$. For subsequent iterations $l > 1$, $\mathfrak{R}_q$ encodes the goal of achieving the termination condition of $a_o^q$, based on its termination condition (Line 4).

Given $\mathfrak{R}_q$, our algorithm learns a policy that maximizes $\mathfrak{R}_q$ using the options available at the current level $l$ (Line 6). For $l = 1$, we assume there is a single option template $\mathcal{A}_o^0 = \{a_o^q\}$; the corresponding policy $\pi_o^q$ aims to achieve the goal in the original environment $\mathcal{E}$, so we refer to the resulting option $o^q = (a_o^q, \pi_o^q)$ as the *top-level option*. For $l > 1$, $\pi_o^q$ implements an option template $a_o^q \in \mathcal{A}_o^{l-1}$ at level $l - 1$ using the option templates available at the current level $l$. Importantly, this process leverages policies learned so far to generate initial states from which to learn $\pi_o^q$. Once we have learned $\pi_o^q$, we add the resulting option $o^q = (a_o^q, \pi_o^q)$ to $\mathcal{D}$.

Recall that the termination condition for an option starting at state $s_0$ is $T(s_0, k^*) = \mathbf{F}_o(s_k) \vee (k > k^*)$. The goal of training $\pi_o^q$ is to satisfy this condition with high probability. Hence, when training policy $\pi_o^q$ for option $a_o^q$, we choose the reward function $\mathfrak{R}_q$ to be $\mathbf{F}_o$, and additionally restrict the length of each episode to $k^*$ time-steps.

**Learning option policies.** Next, we describe LearnOptionPolicy, which Algorithm 2 uses to learn a policy for option template $a_o^q$. For simplicity, we denote the current environment by $\mathcal{E}$, the current reward function by $\mathfrak{R}$, the option template by $a_o$, and the target policy by $\pi_o$. The goal of $\pi_o$ is to achieve the termination condition $T$ for $a_o = (I, T, P)$ from initial states $s \in I$.

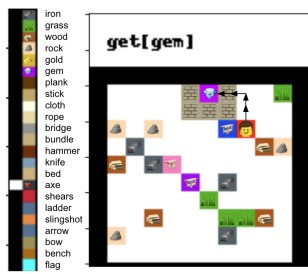
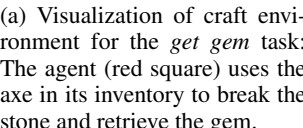

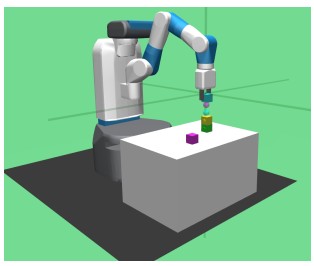

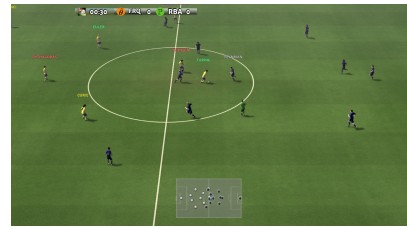

(a) Visualization of craft environment for the *get gem* task: The agent (red square) uses the axe in its inventory to break the stone and retrieve the gem.

(b) Visualization of fetch and stack environment: Each block has to be placed at the location represented by the color-coded sphere.

(c) Visualization of the Google football environment: The agents learn to use 11 players of the left team and score goals.

Figure 3: Craft, Fetch and GFootball environments.

One challenge is sampling an initial state $s \in I$ from which to train $\pi_o$. To do so, this subroutine leverages access to the previously learned options $\mathcal{D}$; it uses options in $\mathcal{D}$ until it arrives at a state $s \in I$ where $a_o$ is called. In more detail, it samples $s$ by executing the top-level option in $\mathcal{D}$ until it reaches a state where $a_o$ is called. In general, executing an option $o$ either relies on executing its policy $\pi_o$ in $\mathcal{D}$ if it exists, or by executing its "teleport" functionality. Our algorithm keeps track of the execution of options for which $\pi_o$ exists using a stack, which is initialized with the top-level option (Line 3-5). The structure is visualized in Figure 2. Then, while executing the current option $o$, it obtains the next option $o'$ to execute, which is processed in one of three ways:

- If $o'$ is the option for $a_o$ (i.e., it has a matching initial state, termination condition, and a current policy $\tilde{\pi}_o$, checked by IsOption on Line 10), then it takes steps to train $\pi_o$ (Line 11); as described below.

- If $o'$ does not have an implementation (checked by HasImplementation on Line 12), then it teleports—i.e., we sample $s \sim P$ (Line 13).

- Otherwise, $o'$ has an implementation, so it pushes $o'$ onto the stack (Line 15).

Finally, once we are at a state $s$ where $a_o$ is called, the subroutine $StepAndTrain$ takes control and trains $\pi_o$ using a standard reinforcement learning algorithm—i.e., by collecting rewards from $s$ and taking a policy gradient step once it reaches a terminal state or hits the timeout of $k^*$ steps. Then, the current episode continues from the state reached. Once the current episode reaches a final state, then the algorithm continues to the next episode. This process continues until the average reward of $\pi_o$ exceeds a threshold $1 - \delta$, or until a maximum number of episodes $L$ is reached.

## 4 Experiments

### 4.1 Experiments on Planning tasks in the Craft Environment

**Description of environment:** The *craft* environment [14, 15] is a 2D world based on the Minecraft game, where an agent has to complete various hierarchical tasks with sparse rewards. The environment is represented by a $12 \times 12$ grid with cells containing raw resources (*e.g., WOOD, IRON*), crafting areas (*e.g., WORKBENCH*), obstacles (*e.g., STONE, WATER*) and valuable items (GOLD or GEM). There are four move actions (*up, down, left, right*) and a special *USE* action. To grab an item, the agent moves to a neighboring cell and applies the *USE* action. Table 1 shows the hierarchical arrangement of the *get gem* task; the details for *get gold* are in the Appendix.

**Option template at each environment:** Our option templates (in Table 1) capture the hierarchy of task-dependencies in the environment. For example, it is easier to get *GEM* when the agent has access to *AXE* (see Table 1). Thus, at the topmost level the agent can "ask" for an *AXE*, and via primitive actions, use it to break stone and get *GEM* to realize the importance of an *AXE*.

**Implementation:** In each task of each learning level, we use a vanilla actor-critic algorithm [16] with a network with one hidden layer of 100 neurons for both the actor and critic. The input to the network is a feature vector consisting of one-hot encodings of the items in each cell in a $5 \times 5$ grid

| Task | Policy Sketch | | Task | Learning level | option templates |
|------|---------------|---|------|----------------|------------------|
| get wood | - | | | | |
| get iron | - | | get gem | 1 | {give axe, primitive actions} |
| make stick | get wood → use anvil | | make axe | 2 | {give stick, give iron, primitive actions} |
| make axe | make stick → get iron | | make stick | 3 | {give wood, primitive actions} |
| | → use workbench | | get iron | 3 | {primitive actions only} |
| get gem | make axe → break stone | | get wood | 4 | {primitive actions only} |

Table 1: Get gem hierarchical task: **[LEFT]** policy sketches [14] & **[RIGHT]** option templates for each task in the hierarchy. The order of the rows represent the learning order in the two alternatives.

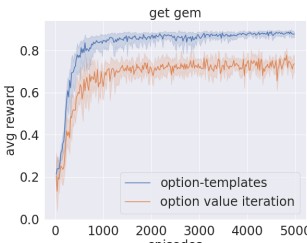 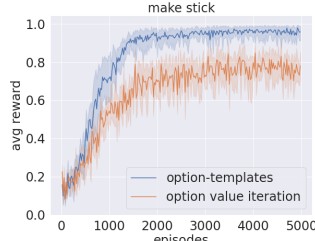 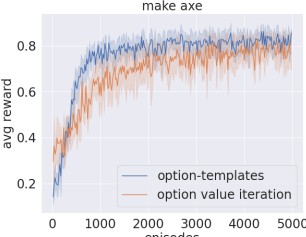

Figure 4: Average reward vs episodes for solving each hierarchical sub-task. We compare option templates (ours) and option-value iteration (baseline) for the task *get gem* in craft environment.

around the agent along with a one-hot encoding of its inventory. At the topmost level, the agent is given a reward of 1 if and only if *GOLD* or *GEM* are obtained. After training, we obtain a actor network for individual tasks at different learning levels.

**Comparison with option-value iteration:** We implement option value iteration [2] as a comparison with standard hierarchical learning. In option value iteration, we learn options bottom-up, using options learnt at a lower level to accomplish sub-tasks of options at the level above. That is the agent learns to implement lower level sub-tasks first before it learns how to use them. We plot average reward as a function of episodes for each sub-task of *get gem* in Figure 4; for *get gold* the Figure is in Appendix A.2. Figure 4 shows that option templates obtain higher average rewards than option-value iteration at all levels (detailed discussion in Appendix A.2).

**Comparison with curriculum learning [14]:**
Additionally, we compare our method with the curriculum learning algorithm employed in Andreas et al. [14] and observe a 100 fold decrease in the number of episodes required to train an agent for the *get gem* and *get gold* tasks (see Table 2 for *get gem* and Appendix for *get gold*). For the proposed method, we report the total episodes it requires for average reward to stay consistently above 0.8. Averaged over ten runs for different random seeds. The calculation of the total episodes for each task, includes all the sub-tasks in its hierarchy. For instance,

| Task | Episodes | |
|------|----------|---|
| | Curriculum learning [14] | Option templates |
| get gem | $> 3 \times 10^6$ | $12826.0 \pm 2613.0$ |
| make axe | $> 2.7 \times 10^6$ | $11283.0 \pm 2255.0$ |
| make stick | $> 1.3 \times 10^6$ | $5026.0 \pm 2231.0$ |

Table 2: Comparison of total episodes (and standard deviations over ten random seeds) to train an agent to solve the *get gem* task via option templates and curriculum learning [14].

the episodes for *get gem* completion via option templates include episodes required for completing *make axe*, *make stick*, *get wood* and *get iron*. The latter two tasks are straightforward, and not included in the table.

## 4.2 Experiments on Manipulation Tasks in the Fetch and Stack Environment

**Description of environment:** This continuous action space environment, introduced in Lanier [17], consists of a robotic arm and a platform with blocks placed on it in random positions; see Figure 3b. The goal is for the arm to move, lift, and release the colored blocks in the location and stacking order specified by the corresponding color-coded spheres. The action consists of torques for 3 degrees-of-freedom actuation, and an on-off control input opens and closes the gripper. The environment offers a sparse reward for each block placed at the goal location.

| Task(s) | level | Option templates |
|---|---|---|
| Fetch & Stack $N$ blocks | 1 | {Place block $i$ at its goal location}$_{i=1,...,N}$ |
| Place block $i$ at its goal location | 2 | {Reach block $i$, Pick block $i$ & reach goal, Release block $i$ & lift, Do nothing}$_{i=1,...,N}$ |

Table 3: Option templates for fetch & stack.

| Task(s) | level | Option templates |
|---|---|---|
| Win game | 1 | {Attack and score goals, Defend} |
| Attack and score goals | 2 | {Maintain ball possession, Charge to the opponent's goal, and Shoot} |

Table 4: Option templates for gfootball.

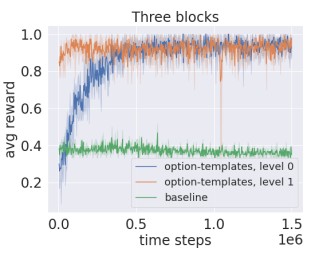
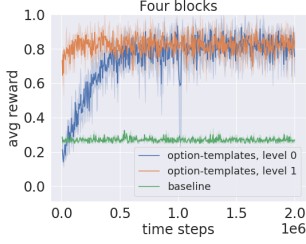
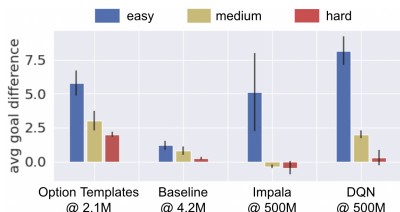

(a) Average reward vs timesteps for option templates and the baseline.

(b) Average reward vs timesteps for option templates and the baseline.

(c) Comparison of time steps and corresponding average goal difference.

Figure 5: Results on the Fetch & Stack and GFootball environments.

**Learning levels and option templates:** We introduce two learning levels with tasks and option templates described in Table 3. In this environment, the primitive actions are *not* continuous-space actions. Instead, at the lowest level (level 2), we expose the agent to options which are implemented with proportional feedback controllers [18]. Such simple primitives have the ability to improve exploration and can be easily transferred among different learning scenarios. Further, the hierarchical structure improves the agent's learning speed as compared to a case where the agent has to learn to use the options at the bottom level (2) directly (see Appendix A.3 for more details).

**Implementation:** We use a standard DQN [12] for each level with four hidden layers of 300 neurons each. The inputs to the agent are the 3D coordinates of the different blocks, their goals, and the states of the gripper arm. The agent has 150 and 200 steps for stacking three and four blocks, respectively, in the correct order. We also supply demonstrations to speed up learning.

**Baseline:** We consider a baseline that directly exposes the agent to all option templates, at level 2 of the hierarchy. Figure 3b shows the average rewards as a function of episodes. As can be seen, the baseline only recieves one-third reward since it only succeeds in pushing the bottom block to its location. It does not learn to stack even after twice the number of learning steps as our method. This demonstrates the challenge in learning longer duration tasks without a higher level guiding policy.

**Comparison with Learning with Demonstrations [1]:** In learning with demonstrations [1], the authors use demonstration traces, by combining behavioral cloning along with a Q-filter mechanism to speed-up the learning. This allows them to evade an expensive exploration phase in the early stages of the learning. This method takes upwards of $3.5 \times 10^8$ and $8 \times 10^8$ timesteps to learn stacking of three and four blocks respectively **which is three orders of magnitude larger than our average learning time** of $4.5 \times 10^5$ and $6 \times 10^5$ timesteps respectively (from 5 random seeds).

### 4.3 Experiments on Multi-Robot Tasks in the GFootball Environment

**Description of environment:** The Google football (gfootball) environment [19] is an 11 vs. 11 game of soccer. The opponents are controlled by an inbuilt game engine with game-play at three levels of difficulty (easy, medium, and hard). An RL agent can control up to 11 players on the left team. The agent provides each player with one of 19 actions such as a direction to move (*e.g.* top, top-right, bottom-left, *etc.*), type of pass (short, long or high), shoot, or toggles for dribbling and sprinting. We provide more details in Appendix A.4.

**Learning levels and option templates:** We create two learning levels with option templates as given in Table 4. The primitive actions in this environment are *defend*, *maintain ball possession*, *charge to the opponent's goal* and *shoot* which are options implemented with simple planers and open-loop controllers. Similarly, we implement option templates at level 1 (see Appendix A.4).

**Implementation:** We train a standard DQN [12] for each level. The input to every network is a 139 element vector consisting of left and right team states, ball state, score, and one-hot encodings of ball ownership and game mode. For level 1, for all levels of difficulty, we use a network with 2 hidden layers of 500 neurons each. For level 2, we use a network with 5, 6, and 7 hidden layers for easy, medium and hard respectively where each hidden layer has 500 neurons.

**Baseline:** We consider a baseline where the agent is directly exposed to all the primitive options. From Figure 5c, we find that the baseline is unable to compete with option template learning even after twice the number of steps of our method (numerical values of Figure 5c are in Appendix A.4). The error bars represent standard deviation over 5 random seeds.

**Comparison with IMPALA and DQN from Kurach et al. [19]:** We compare our agent with the agents in Kurach et al. [19]. As can be seen in Figure 5c, option template learning can achieve similar (in easy) or better (in medium and hard) performance than the IMAPALA and DQN agents in two orders of magnitude fewer steps.

## 5 Related Work

We present a detailed discussion on related work in Appendix A.1 and summarize closely relevant literature here.

**On expert help via primitives or skills**: Various recent works utilize expert help in the form of primitives or skills [4, 5, 6, 7, 8, 9, 20]. This takes place via parameterized action spaces [4], stitching together independent task schemas (or skills) [5, 6, 9, 20] or learning parameters of action primitves [7, 9, 8]. In all of these cases, learning takes place within the traditional hierarchical framework, *i.e.*, bottom-up (see Figure 2), where a sub-task is learnt before the policy that uses it. Our method proposes a framework to learn top-down instead with large-improvements in sample-efficiency over traditional bottom-up learning. With minimal changes in the implementation of their primitives and skills, the above diverse strategies can also benefit with shorter learning times by utilizing our framework to learn top-down.

**On expert help with humans-in-the-loop**: Other approaches to expert intervention include explicit help with humans-in-the-loop required throughout training [21, 22]. In contrast, in our method, human effort is only required at the beginning to create the option template hierarchies.

**On exploration in high-dimensional tasks**: An alternative approach to counter the need for exploration in high dimensional tasks is through the use of demonstrations [1] for long horizon planning. The intuition is that introducing a degree of *behavioral cloning* of expert demonstrations helps reduce the amount of exploration the agent has to perform. A different approach known as *Hindsight Experience Replay* (HER) [23] incorporates the goal information into the state, using a failed terminal state as an alternative goal to reward the transitions leading to it. These methods are orthogonal to our approach. By themselves, they are unable to achieve the large degree of reduction in sample-complexity seen with our framework. Yet, alongside expert help, they further improve sample-efficiency as seen in our experiments.

## 6 Limitations and Conclusion

We have proposed an approach that incorporates *option templates* into reinforcement learning. Our experiments show that this strategy can drastically reduce sample complexity by implementing teleportation. This can be a potential limitation. In simulation, implementing teleportation is typically straightforward. Policies may be trained in simulation before being deployed on a real robot. For real-world environments, there are several strategies for implementing teleportation. First, for challenging skills such as grasping an object, teleportation can be implemented via a temporary crutch that simplifies the skill. For example, in the "sticky-mittens" experiments (a key motivation for our work); the analog for a grasping robot would be to attach velcro to its grippers and to the objects to make them easy to pick up. Second, teleportation can be implemented via a handcrafted policy that eventually achieves the goal but possibly in a suboptimal way. For instance, teleportation in the Fetch and GFootball environments are implemented using handcrafted policies. While these policies are used to train the RL policy, the RL policy eventually significantly outperforms them.

## Acknowledgements

This work was supported in part by ARO W911NF-20-1-0080 and AFRL and DARPA FA8750-18-C-0090. Any opinions, findings, conclusions or recommendations expressed in this material are those of the authors and do not necessarily reflect the views of the Air Force Research Laboratory (AFRL), the Army Research Office (ARO), the Defense Advanced Research Projects Agency (DARPA), the Department of Defense, or the United States Government.

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
