# OpenReview forum: "Exploring with Sticky Mittens: Reinforcement Learning with Expert Interventions via Option Templates"
_robot-learning.org/CoRL/2022/Conference — CoRL 2022 Poster_

### Official Review · Reviewer_XdN5 · 2022-07-28

**Originality:** Fair
**Technical Quality:** Good
**Clarity Of Presentation:** Poor
**Impact:** 3

**Recommendation:**

Weak Accept: I recommend accepting the paper, but will not argue for my recommendation if the majority of other reviewers have a different opinion.

**Summary:**

The paper "Exploring with Sticky Mittens: Reinforcement Learning with Expert Interventions via Option Templates" introduces a hierarchical reinforcement learning approach that divides long time-scales in multiple "option templates", i.e. a group of initial and terminal states without option-specific, low-level policies. The low-level control policies then only come into play at the very end, to successively implement the previously found option templates. The approach is evaluated on multiple reinforcement learning benchmarks in simulation.

**Issues:**

* Clarity: Some of the mathematical definitions are very vague (definition of environment at learning level l, the definition of the reward functions, the satisfaction of the termination condition). In addition, some things remain undefined, such as "teleporting" or how exactly the policies are optimized. In addition, despite having a section "Learning option policies", from the experiments it seems that some option-policies are given (Tables 1, 3 and 4). Furthermore, the "option templates" shown in these tables correspond to policies rather than "option templates" (which should only be defined over sets of states as defined in the paper). Lastly, it remains unclear, how exactly these option templates are defined. In the paper, it says that "For l = 1, we assume there is a single option template A0", however, what precisely is this option template and how is the division over time for subsequent layers? To summarize, the general concept is clear, but the detailed explanations are really confusing and even seem contradictory to some degree. The respective sections really have to be improved upon clarity and precise definitions. The rather involved algorithms do not bring any additional benefit in this case. Maybe the space could be used to leave the respective sections more space.

* Technical Novely: The additional benefit of the proposed approach, especially compared to [1], has to be put more in focus.

* Evaluation: The proposed method claims to yield better sample complexity due to the limited horizon of each subtask by leveraging option templates in a hierarchy. However, the experiments do not really analyze the effect of this hierarchy. How does granularity (i.e. the number of layers) influence performance, which kind of division does bring the most benefit? As is, the experiments only show results for two levels (with the first being the normal task, it seems) which is not really convincing regarding the general idea.

[1] Learning Multi-Level Hierarchies with Hindsight, Levy et al., ICLR 2019

-------------------------------------

Post-rebuttal:
While I appreciate the rebuttal, the paragraph about the effect of the hierarchy is not really convincing. One key aspect of the approach, splitting a task into multiple subtasks to account for long horizons, is not well analyzed, so the experimental evaluation is really lacking some general insights beyond the chosen simulated benchmarks. On the flip side, the results themselves are impressive and hint to at least some real-life applicability -- so it really depends on the scope of CoRL whether this manuscript would be a candidate for acceptance. However, if the camera ready improves upon clarity and presentation, I rather tend to raise my score to WA.

**Quality Of The Limitations Section:**

Additional details required

**Reviewer Expertise:**

4: The reviewer is confident but not absolutely certain that the evaluation is correct

**Robotics Focus:**

Highly relevant to robotics but no hardware experiments

**Strengths And Weaknesses:**

Strengths:

	* In general, the research topic of the submitted manuscript is of high practical relevance and hence of interest to the community of CoRL.
	* It is well motivated and the background is well presented.

Weaknesses:

	Unfortunately, the manuscript in its current form comes with a couple of issues.

	* Clarity: The explanation of the approach is very confusing and not easy to grasp for external readers.
	* Technical Novely: The general idea seems very similar to previous work, e.g. [1]. In addition to the lack of clarity, the contributions to the field remain vague and seem minor.
	* Evaluation: Even though the shown results in the given benchmarks are somewhat convincing, they don't really highlight the special ingredients of the proposed method.

[1] Learning Multi-Level Hierarchies with Hindsight, Levy et al., ICLR 2019

**Summary Of Recommendation:**

The current manuscript neither explains the concept very well, nor does it really convince regarding real-life impact. Despite the general idea being very suitable for CoRL, this paper is no candidate for acceptance in its current form.

---

> ### Author Response · Authors · 2022-08-21
> **Response to reviewer XdN5 (Part 1 of 2)**
>
> We thank reviewer XdN5 for the efforts put into their review. We are glad to see that the reviewer thinks our work is of good technical quality.
>
> # Environment and reward function at each level.
> The environment at each level l has its action space replaced by the option templates of the level. For example, in craft, with the get gem task, as shown in Table 1, the action space for the environment at level 1 is {give axe option template, primitive actions}. The action space for the environment at level 2 is {give stick option template, give iron option template, primitive actions}. This is formalized in Definition 3.2 of lines 121-122. Further, only level 1 (the topmost level) in all environments uses the original (sparse) reward function of the environment. For all levels below the topmost level, the reward function takes a value 1 when the termination condition of the option template being learnt is reached and 0 otherwise. For example, for level 2 in Table 1, only when the make axe option template’s termination condition is reached (i.e. when an axe has been made), reward takes a value of 1. It is 0 otherwise. This is formalized in lines 134-140.
>
> # "Teleporting" is undefined.
> An option template consists of initial states, a terminating condition, and a distribution P of state for which the terminating condition is satisfied. Teleportation, as mentioned in Line 168, is simply the process of sampling from P, i.e. jumping to a final state of an option template, from an initial state. In simulation, teleportation is implemented via simple modifications to the environment (Craft) or via handcrafted policies (Fetch & Stack, G Football). We discuss some strategies to implement teleportation in the real-world in lines 285-293.
>
> # Despite having a section "Learning option policies", it seems that the policies are given in Tables 1, 3 and 4.
> In fact, Tables 1, 3, and 4 only show option templates; indeed, they correspond to sets of states as the reviewer suggests, and not policies. Specifically, Tables 1, 3 and 4 provide descriptive names for all option templates based on their termination condition. For example, in Craft, give axe option template is named because it adds an axe to the inventory of the agent. Its terminal condition is when a new axe is available in the agent’s inventory. In Fetch & Stack, the option template named place block $i$ at its goal location terminates when block $i$ is placed at its goal location. To avoid confusion, we can update these descriptions to be more declarative, e.g., axe in inventory for Craft and block $i$ is at its goal location for Fetch & Stack.
>
> In both the Craft and Fetch & Stack environments, the initial states of all option templates are all possible states. In the G Football environment, we describe the initial states and termination conditions in the Appendix in lines 476-499.
>
> Further, the timeout for option templates in Craft is given in lines 427-429 of the Appendix, for Fetch & Stack in lines 454-455 of the Appendix, and for GFootball in line 500 of the Appendix.
>
> Our algorithm in the section on ‘learning option policies’ explains how to learn a Deep RL based policy for each option template (option template + Deep RL policy = option).
>
> # What is the one option template at level 1?
> This is the complete task for which we are learning a policy. We used this formalism to simplify exposition, since there was no need to introduce the final task as a separate learning problem. In Craft, that can be get gem or get gold. For Fetch and Stack, it is Fetch and Stack N blocks where N can be 3 or 4. For G Football, it is Win game.
>
> We provide a level-by-level explanation of Craft here. In Craft’s get gem task, in level 1, the policy of the get gem option template is learnt with an action space given by {give axe option template, primitive actions}. The give axe option template adds an axe to the agent’s inventory when used.
>
> The policy of get gem learnt above in level 1 is executed until a state where give axe is called. Starting from this state, the policy of the make axe option template is learnt in an action space given by {give stick, give iron, primitive actions}. The give stick and give iron option templates add a stick and axe respectively to the agent’s inventory.
>
> This process is repeated for all levels.
>
> After training, we have a policy (in this case, an actor and critic neural network) for each option template. To test our policies, we compose these neural networks replacing the give X option template with the corresponding make X implementation at each level.
>
> The action space for learning encodes what lower level option templates are available for learning the policy of an higher level option template.

---

> ### Author Response · Authors · 2022-08-21
> **Response to reviewer XdN5 (Part 2 of 2).**
>
> # Effect of the hierarchy.
> Even though this is important when training policies for a given environment in general, we focus on the process of learning for a fixed hierarchy, that is, going bottom-up vs top-down. Some of the hierarchies we used in our experiments come as part of the task specification from the environment (Craft environment). We show that across different descriptions of the hierarchy in different settings option-template guided top-down approaches can provide a practical algorithm that can learn a policy in a realistic amount of training steps.
>
>
> # Benefits when compared to ‘Learning multi-level hierarchies with hindsight [Levy et al 2019]’
> The current paper is distinct in the high-level approach compared to [Levy et al 2019], in the following ways :
>
> * In [Levy et al 2019], the transitions on taking an action at a certain level, depend on the policy at the immediate level below. But, these transitions have the problem of being non-stationary due to the updates to the lower level policy. The authors circumvent this issue by using hindsight action transitions, which uses an intermediate state reached at the lower-level as a possible action in the current level. For the approach proposed in this paper, we do not need to rely on the performance of lower level policies to guide the exploration of higher level goals. We provide jumps in the state-space which are expert informed in the form of option-templates. The policy at the higher-level learns how to combine the option-templates such that it can achieve the goal. In complex tasks like Fetch and Stack, it is unlikely that, left to random exploration, the agent will learn to grasp an object in the first place.
>
> * Another point of distinction between the two approaches, is that the higher-level policies iteratively changes the sub-goals to be achieved by the lower-level policy. This is not the case in our implementation where the two stages are decoupled. The definition of an option-template remains static. A policy using an option-template is free to invoke it at any point in the learning algorithm.
>
> We also trained and tested [Levy et al 2019]’s Hindsight Actor Critic (HAC) algorithm on Fetch & Stack environment with three blocks and clearly notice that it cannot learn to stack two blocks in more than 10 times our learning time.
>
> **HAC [Levy et al 2019] obtain an average reward of only 0.138 after $(67.2 \pm 0.1) \times 10^5$ steps (obtained over 5 random seeds). In comparison, we obtain an average reward of 1 after a learning duration of only $(4.5 \pm 0.1) \times 10^5$ timesteps.**
>
> Note that, we use the high-level / low-level learning strategy described in [Yang et al 2020] and only mention the high-level learning time (which is much lower than the low-level learning time) for [Levy et al 2019] here for a fair comparison. **We plot the variation of rewards at all levels with our method, baseline and HAC [Levy et al 2019] in the attached document.**
>
> Further, the method proposed in [Yang et al 2020] called Universal Option Framework claims to outperform [Levy et al 2019]’s Hindsight Actor Critic. Even with this boost, according to [Yang et al 2020], their method attains a 0.7 average reward at the high-level only after $(480 \pm 3.2) \times 10^5$ timesteps.
>
> [Levy et al 2019] Learning multi-level hierarchies with hindsight. ICLR 2019.
>
> [Yang et al 2020] Hierarchical Reinforcement Learning With Universal Policies for Multistep Robotic Manipulation. IEEE Transactions on Neural Networks and Learning Sytems.

---

### Official Review · Reviewer_dp9v · 2022-08-01

**Originality:** Good
**Technical Quality:** Good
**Clarity Of Presentation:** Very Good
**Impact:** 4

**Recommendation:**

Weak Accept: I recommend accepting the paper, but will not argue for my recommendation if the majority of other reviewers have a different opinion.

**Summary:**

The paper provides an interesting perspective to existing Option-based hierarchical RL methods by proposing the idea of Option Templates: It introduces a novel framework to learn option templates, which enables the agent to have expert intervention as high-level option of the task when they are unsure how to continue the action from the state. Specifically, it could transit from one state to another immediately, where the transition represent the execution of an unimplemented skill primitive. It lays out different levels of option templates, and eventually learns the lower level actions in the environment. This could improve the policy performance and sample efficiency of RL and the paper shows the effectiveness of this method on tasks including robot manipulation and soccer.

**Issues:**

As discussed in the Weaknesses section above:

1. There should be more discussion on how the expert intervention can be obtained, and the cases where it is unable to obtain a high-performing policy as the expert.

2. There should be experiments on more complex tasks like more challenging manipulation of objects (e.g. grasping) instead of pushing and moving objects. Also the skills should be more realistic than simply placing one block at a certain location. If there are limitations to doing such experiments, they should be addressed in the limitation section.

3. More reference and comparison need to be made to intervention-based / interactive / human-in-the-loop RL


**Quality Of The Limitations Section:**

Additional details required

**Reviewer Expertise:**

3: The reviewer is fairly confident that the evaluation is correct

**Robotics Focus:**

Relevant but unlikely to deploy to hardware in near future

**Strengths And Weaknesses:**

Strengths:
- The motivation for using Option Template from the idea of Sticky Mitten is interesting and strong. It provides a nice analogy between human and agent learning.

- It is a very practical and interesting idea to have expert interventions as high-level instructions which largely reduces the exploration burden and provide learning signals to the agent.

- The explanation on option template method and idea is very clear. There are clear comparison between the option template

- The limitation section provides true and to-the-point discussion on the sources of teleportation, which should be considered in future work.

- There are extensive discussions on how this approach compared with prior methods like option-value iteration and curriculum learning.

Weaknesses:

- The main concern of the paper is how well it could be applied to more complex robotics tasks that are 1) have longer-horizon which motivates the use of hierarchical RL 2) where the skills are not clearly defined. The robot experiments in the paper involves simple moving of objects, and does not reflect how hierarchical RL could help with more complex tasks.

- There are no real robot experiments and no discussions of how well this could be scaled to real-world setting where there are more intricate manipulation of objects and more complex skills.

- For more complex, long-horizon robotics tasks, how could the skills be extracted and the options to be formed in the first place? Does it assume that it already have skill segmentation done? The learning levels here seems a bit contrived and manually designed for each individual task.

- While comparing the Option Template method to previous hierarchical RL method is sound, the idea of Option Template have the use of expert intervention, which relates to the line of work on learning with expert interventions. While this is a different formation and a different framing, there should be references made to these methods, listed a few in below:
1. Efficient Learning of Safe Driving Policy via Human-AI Copilot Optimization [Li 2022]

    https://arxiv.org/abs/2202.10341
2. Expert Intervention Learning [Spencer 2020]

    https://link.springer.com/article/10.1007/s10514-021-10006-9



**Summary Of Recommendation:**

I think the Option Template is a very interesting idea that complements the existing option framework and addresses the limitations in previous hierarchical reinforcement learning methods. While it is in theory an interesting approach, the experiments in the paper haven't been able to show the effectiveness of the approach in complex robot manipulation tasks. The fact that the tasks are mainly in grid world and the robotics tasks are simply pushing and placing objects are a bit limited. The authors need to show more evidence on how this method could be used on more complex tasks like robot manipulation etc. Also, there should be more evidence of how this could be scaled to real-world setting with real robots.

---

> ### Author Response · Authors · 2022-08-21
> **Response to reviewer dp9v.**
>
> We thank reviewer dp9v for their detailed review and efforts put into the same. We are glad to see that the reviewer appreciates our paper and the impact is major.
>
> # Experiments on more complex tasks like more challenging manipulation of objects (e.g. grasping)
> We have performed experiments in Fetch and Stack environment. This involves fetching 4 different colored blocks (by grasping with a two-finger robot claw) and vertically stacking them in the right order. An example of the 4 block stack that has to be created in the task is shown mid-task in Figure 3 (b). We are not limited to just simple moving of objects. Further, we also tackle the challenging multi-agent control task (11-players) in the G Football environment. In G-Football, the composition of option-templates are non-trivial, and require interesting orchestration of actions to score a goal. We believe this framework is thus general given the range of environments it appears to perform well in. Finally, we emphasize that our work is designed to address challenges in high-level, long-horizon planning problems rather than complex low-level control problems such as dexterous manipulation.
>
> # Does it assume that it already has the skill segmentation?
> Yes, we assume it is possible to decompose the tasks into an option template hierarchy. The lower-level primitives make it practically feasible to learn a policy in a reasonable amount of time.  Creating such a hierarchy for the three complex environments -- Craft, Fetch and Stack and G Football, we found was straightforward. We believe our approach provides a reasonable tradeoff between the amount of knowledge provided without imposing undue burden on the designer.
>
> # Reference to [Li 2020] and [Spencer 2020]
> Thank you for this interesting recommendation. We will mention both HACO [Li 2020] and EIL [Spencer 2020] as alternate approaches for leveraging expert intervention, but with humans-in-the-loop required throughout training. In contrast, in our method, human effort is only required at the beginning to create the option template hierarchies.
>
> # There should be more discussion on how the expert intervention can be obtained
> For simulators, expert intervention (in our case, the option templates or teleportation) can be obtained via straightforward modifications to the simulator (such as in Craft) or via handcrafted policies (such as in Fetch & Stack and G Football). Moreover, if a handcrafted policy is used, it does not have to be a high-performing policy. In fact, a sub-optimal policy, which takes a larger number of time steps, is fine as well. As long as it can implement a jump in the state-space, any sub-optimal handcrafted policy can be used.
>
> Further, the option template hierarchy can be created for some environments based on the task descriptions itself. For example, in Craft, the policy sketches [14] available in the environment directly help create the option templates in Table 1. We were also inspired by [1]’s description of the sequence of actions to stack blocks to create our option templates in Fetch & Stack (Table 2) and by [19] for G Football (Table 3). We describe the option templates in G Football and the reasoning behind them in lines 476-499 in the Appendix.
>
> For real-world tasks, expert help (teleporation) can be provided in three ways. First, by training policies in simulation (as described above) and transfering to the real world. Second, by using sub-optimal handcrafted policies (as described above). Third, by using a temporary crutch that simplifies the task. For example, in the “sticky-mittens” experiments (a key motivation for our work); the analog for a grasping robot would be to attach velcro to its grippers and to the objects to make them easy to pick up. We describe this in Lines 284-293.

---

### Official Review · Reviewer_zrnS · 2022-08-01

**Originality:** Very Good
**Technical Quality:** Very Good
**Clarity Of Presentation:** Good
**Impact:** 4

**Recommendation:**

Weak Accept: I recommend accepting the paper, but will not argue for my recommendation if the majority of other reviewers have a different opinion.

**Summary:**

The main contribution of this paper is a sample-efficient RL approach for learning policies for long-horizon tasks, by making use of multiple levels of abstraction, each of which is associated with a set of option templates. An option template defines an option, in terms of initial and final states, but not how to implement it. All option templates are handcrafted at the start of training, so the agent can execute an option template despite not knowing how to implement it yet. During training, the agent learns how to implement each option template in terms of the option templates at the next lower level of abstraction.

The paper evaluates this approach in three domains, all in simulation: craft (a grid world based on Minecraft), stacking with a robot arm, and 11 vs. 11 football. This top-down approach of learning options reaches higher performance than learning the options first and then combining them. It is orders of magnitude more sample-efficient than learning without options.

**Issues:**

### Suggestions / questions:
* Points 3, 4, and 5 from Weaknesses section above
* I'm not sure what Figure 2 is trying to communicate. The graph structure is confusing to me, since each option is executed by a _sequence_ of options at the next level, so ordering matters. I think it would be useful to make Figure 2 more concrete, as suggested above.
* End of Section 3: What happens if the maximum number of episodes L is reached but the option has not been learned properly yet?
* It is worth making space for Related Work in the main paper, rather than leaving it to the Appendix. I recommend mentioning Task and Motion Planning (TAMP) in Related Work, as an alternate hierarchical approach for tackling long-horizon problems in robotics. e.g. Garrett et al. Integrated Task and Motion Planning. https://arxiv.org/abs/2010.01083

### Typos / wording:
* Page 1, line 20: "Designed to achieve intermediate subgoals." - not a complete sentence
* in Algorithm 2, in line 6 shouldn't the initial state be sampled from $\mathcal{I}$, rather than $I$? (i.e., the initial states of the semi-MDP rather than of the option)
* Page 5, line 173: Missing a space before "Once"
* Page 8, line 255: "planers" → "planners"

**Quality Of The Limitations Section:**

Limitations are addressed clearly

**Reviewer Expertise:**

3: The reviewer is fairly confident that the evaluation is correct

**Robotics Focus:**

Highly relevant to robotics but no hardware experiments

**Strengths And Weaknesses:**

### Strengths:
* The proposed approach demonstrates orders of magnitude improvement in sample-efficiency, compared to state-of-the-art RL approaches that do not use options.
* The approach is evaluated in complex continuous control domains.
* Includes an interesting comparison that shows this approach of learning top-down with option templates works better than learning bottom-up (i.e., learning the options first and then connecting them together). The hypothesis is that this is because the learned options typically do not perform perfectly, whereas option templates do, which makes it easier to learn how to correctly connect options together when using the latter.
* Clear explanation of the background, in terms of options and semi-MDPs.

### Weaknesses:
* The approach requires defining the levels of abstraction and the set of option templates in each. The effectiveness of the approach presumably depends greatly on how well the problem is decomposed into these levels of abstraction and option templates.
* The implementation of option templates can be nontrivial, as is acknowledged in the Limitations section.
* The description of the method could be more clear. Including a more detailed Figure 2 would help, perhaps with more concrete examples of what the options could be (e.g., from the Minecraft task), and showing how over the course of training, each of the options is learned in terms of the next level of options. Also, in the implementation of the approach, how does one restrict the policy to pick from the set of options that are valid in a particular state?
* There is no comparison against option value iteration in the manipulation and football environments. Is it because option value iteration would take too long to learn for these tasks?
* No evaluation on real robots. This would be a stand-out paper for CoRL if the evaluation included learning a long-horizon task with a real robot.

**Summary Of Recommendation:**

This paper proposes an exciting new approach of using options in a top-down manner, that could be highly effective at learning long-horizon tasks efficiently on real robots. This approach does require significantly more effort to apply than end-to-end RL, since the abstraction levels and sets of option templates need to be defined, as well as a handcrafted policy for each option template. But this could also be an advantage, in that it enables combining low-level controllers with higher-level learned policies. With a few improvements in clarity and evaluation, I think this would be a strong paper and very relevant for the CoRL community.

---

> ### Author Response · Authors · 2022-08-21
> **Response to reviewer zrnS.**
>
> We thank reviewer zrnS for their detailed review and efforts put into the same. We are glad to see that the reviewer supports our work.
>
> # More concrete examples of what the options could be (e.g., from the Minecraft task). Restricting the policy to pick from the set of options that are valid in a particular state.
> We take a deep-dive with the Craft environment here. In Craft’s get gem task, in level 1, the policy of the get gem option template is learnt with an action space given by {give axe option template, primitive actions}. The give axe option template adds an axe to the agent’s inventory when used.
>
> The policy of get gem learnt above in level 1 is executed until a state where give axe is called. Starting from this state, the policy of the make axe option template is learnt in an action space given by {give stick, give iron, primitive actions}. The give stick and give iron option templates add a stick and axe respectively to the agent’s inventory.
>
> This process is repeated for all levels.
>
> After training, we have a policy (in this case, an actor and critic neural network) for each option template. To test our policies, we compose these neural networks replacing the give X option template with the corresponding make X implementation at each level.
>
> The action space for learning encodes what lower level option templates are available for learning the policy of an higher level option template.
>
> # Explanation of Figure 2.
> Figure 2 is a visual representation of the order in which learning happens. The option-templates are used in the current level as actions to achieve the goal. These option-templates become the sub-tasks for the following level. This continues until the lowest level actions are reached.
>
> # No comparison against option value iteration in the manipulation and football environments.
> The bottom-up mode of learning in the Fetch and Stack and G Football environments is implemented using option-value iteration. This is straightforward to implement since, if the lower level options are available, then the higher level policy can simply use these while performing option-value iteration.
>
> # I recommend mentioning Task and Motion Planning (TAMP) Survey in Related Work.
> Thank you for this interesting recommendation. We will mention TAMP methods as alternate hierarchical approaches for tackling long-horizon problems in robotics. Generally, TAMP approaches highlight the interplay of motion-level and task-level planning, with the task-level planner constraining the motion-level planner. In contrast, our approach allows the task-level planner to learn using feasible paths which can be satisfied by the motion-level planner. This allows us to achieve high reward with few samples.
>
> # What happens if the maximum number of episodes L is reached but the option has not been learned properly yet?
> L is a hyperparameter. We increase L if the option has not been learned yet.

---

### Author Response · Authors · 2022-08-21
**Thank you to Reviewers and Area Chair**

We thank all three reviewers and the area chair for their detailed reviews and efforts put into the same. We are glad that reviewers zrnS and dp9v think our work is high impact, and our originality and clarity is good or very good. We are happy to see that all reviewers think our paper has good or very good technical quality.

---

### Meta-Review · Area_Chair_6pYK · 2022-08-12

**Recommendation:** Accept (Poster)
**Confidence:** 3

**Metareview:**

The strength and weakness of the paper raised by reviewers are summarized as follows:
Strength:
- The motivation and ideas are clearly presented
- The approach is evaluated on complex continuous control domains such as fetch-and-stack and G Football tasks, and the benefits of the proposed approach is apprppriately demonstrated.
- Empirical comparison between the bottom-up and top-down approaches is interesting

Weakness:
- The proposed approach requires the option templates, which is not trivial to extract. It is not clear how to extract the option templates for tasks with longer horizons.
- More in-depth discussions on the related work are necessary

In addition to the weakness raised by reviewers, AE raises the following concerns:
- the comparison with IMPALA, DQN and the method in A. Nair et al. [1] in fetch-and-stack and GFootball tasks does not look fair. The task is significantly simplified for the proposed method using the pre-defined option templates, but it seems that the option templates are not given to baseline methods.
- It seems that there is a gap between the motivation in the introduction and what is done in the proposed method. From the introduction, AE thought that all the layers would be learned simultaneously in the proposed method. However, in the proposed method, the lowest level policy is manually designed and given to the agent as option templates.

Overall, the paper contains interesting ideas and results. However, it is necessary to improve the presentation in some parts.

=== post-rebuttal comments ===

While some concerns were raised by reviewers, the authors addressed them by deepening the discussion and providing additional results.  Considering that reviewers agree that the paper presents interesting ideas and results, AE recommend the acceptance of the paper. AE also encourages the authors to check the comments from the reviewers once again and make some more efforts to improve the presentation.

**Best Paper Nomination:**

No

---

> ### Author Response · Authors · 2022-08-21
> **Response to Area Chair (Part 1 of 2).**
>
> # Not trivial to extract option templates.
> In our paper, we specify the option templates for an environment before learning the policy for each option template. For the environments in our experiments, we found that specifying option templates was reasonably straightforward. For instance, for the Craft environment, these come in the form of well separated sub-tasks such as, “build bridge to cross water”.
>
> Moreover, existing RL algorithms are impractical in realistic settings due to their high sample complexity. It is nearly impossible to build a general-purpose RL algorithm that achieves good sample complexity due to the inherent intractability of the corresponding optimization problems. Incorporating prior knowledge provides a way for experts to guide the learning problem and reduce sample complexity. We believe our approach provides a reasonable tradeoff between the amount of knowledge provided (specification of the possible options that can be used) without imposing undue burden on the designer.
>
> # Comparison with IMPALA, DQN in [19] and Learning with Demonstrations [1].
> We provide the results with these three methods simply as a reference point for the number of training steps and corresponding reward achieved by the state-of-the-art in the Fetch and Stack and G Football environments respectively. Our baselines are bottom-up methods that are given the option templates for a fair comparison with our method (which is top-down). The baseline in fetch and stack is unable to learn to stack (it can only push the bottommost block to its position). In G Football, the baseline achieves a very low reward even after taking twice as many steps as our method.
>
> # The lowest level policies are manually designed and given to the agent.
> In Fetch and Stack, and G Football, the lowest level options are given (as simple proportional feedback and open loop controllers, named primitives or skills, that can be transferred among tasks), but in Craft, all levels are learnt including the lowest level. Moreover, even after providing the same lowest level options to the baselines (bottom-up methods), they completely fail to learn, even after 2x the training steps. The top-down approach is a clear winner when compared to the bottom-up approach. Moreover, various recent works [4, 5, 6, 7, 8, 9, 20] utilize expert help in the form of primitives or skills and they do so within the bottom-up framework. Our work suggests that these papers using primitives or skills could significantly benefit from a top-down approach.
>
> # More discussion on related work.
> We will mention the TAMP survey [Garett et al 2020] as alternate hierarchical approaches for tackling long-horizon problems in robotics as requested by Reviewer zrnS. Generally, TAMP approaches highlight the interplay of motion-level and task-level planning. The task-level planner constrains the motion-level planner. What separates our approach, is that we allow the task-level planner to learn using feasible paths which can be satisfied by the motion-level planner. This allows us to achieve high reward with few samples.
> We will mention both HACO [Li 2020] and EIL [Spencer 2020] as alternate approaches for leveraging expert intervention (as requested by Reviewer dp9v), but with humans-in-the-loop required throughout training. In contrast, in our method, human effort is only required at the beginning to create the option template hierarchies.  (This discussion can be found in the responses to Reviewers zrnS and dp9v as well.)

---

> > ### Author Response · Authors · 2022-08-22
> > **Response to Area Chair (Part 2 of 2)**
> >
> > # Some more discussion on related work: benefits when compared to ‘Learning multi-level hierarchies with hindsight [Levy et al 2019]’
> > The current paper is distinct in the high-level approach compared to [Levy et al 2019], which was requested by Reviewer XdN5, in the following ways. (This discussion can be found in the response to Reviewer XdN5 as well.)
> >
> > * In [Levy et al 2019], the transitions on taking an action at a certain level, depend on the policy at the immediate level below. But, these transitions have the problem of being non-stationary due to the updates to the lower level policy. The authors circumvent this issue by using hindsight action transitions, which uses an intermediate state reached at the lower-level as a possible action in the current level. For the approach proposed in this paper, we do not need to rely on the performance of lower level policies to guide the exploration of higher level goals. We provide jumps in the state-space which are expert informed in the form of option-templates. The policy at the higher-level learns how to combine the option-templates such that it can achieve the goal. In complex tasks like Fetch and Stack, it is unlikely that, left to random exploration, the agent will learn to grasp an object in the first place.
> >
> > * Another point of distinction between the two approaches, is that the higher-level policies iteratively changes the sub-goals to be achieved by the lower-level policy. This is not the case in our implementation where the two stages are decoupled. The definition of an option-template remains static. A policy using an option-template is free to invoke it at any point in the learning algorithm.
> >
> > We also trained and tested [Levy et al 2019]’s Hindsight Actor Critic (HAC) algorithm on Fetch & Stack environment with three blocks and clearly notice that it cannot learn to stack two blocks in more than 10 times our learning time.
> >
> > **HAC [Levy et al 2019] obtain an average reward of only 0.138 after $(67.2 \pm 0.1) \times 10^5$ steps (obtained over 5 random seeds). In comparison, we obtain an average reward of 1 after a learning duration of only $(4.5 \pm 0.1) \times 10^5$ timesteps.**
> >
> > Note that, we use the high-level / low-level learning strategy described in [Yang et al 2020] and only mention the high-level learning time (which is much lower than the low-level learning time) for [Levy et al 2019] here for a fair comparison. **We plot the variation of rewards at all levels with our method, baseline and HAC [Levy et al 2019] in the attached document.**
> >
> > Further, the method proposed in [Yang et al 2020] called Universal Option Framework claims to outperform [Levy et al 2019]’s Hindsight Actor Critic. Even with this boost, according to [Yang et al 2020], their method attains a 0.7 average reward at the high-level only after $(480 \pm 3.2) \times 10^5$ timesteps.
> >
> > [Levy et al 2019] Learning multi-level hierarchies with hindsight. ICLR 2019.
> >
> > [Yang et al 2020] Hierarchical Reinforcement Learning With Universal Policies for Multistep Robotic Manipulation. IEEE Transactions on Neural Networks and Learning Sytems.